# Iterative point set registration for aligning scRNA-seq data

**Amir Alavi**[1], **Ziv Bar-Joseph**[1,2]*

**1** Computational Biology Department, School of Computer Science, Carnegie Mellon University, Pittsburgh, Pennsylvania, United States of America, **2** Machine Learning Department, School of Computer Science, Carnegie Mellon University, Pittsburgh, Pennsylvania, United States of America

* zivbj@cs.cmu.edu

## Abstract

Several studies profile similar single cell RNA-Seq (scRNA-Seq) data using different technologies and platforms. A number of alignment methods have been developed to enable the integration and comparison of scRNA-Seq data from such studies. While each performs well on some of the datasets, to date no method was able to both perform the alignment using the original expression space and generalize to new data. To enable such analysis we developed Single Cell Iterative Point set Registration (SCIPR) which extends methods that were successfully applied to align image data to scRNA-Seq. We discuss the required changes needed, the resulting optimization function, and algorithms for learning a transformation function for aligning data. We tested SCIPR on several scRNA-Seq datasets. As we show it successfully aligns data from several different cell types, improving upon prior methods proposed for this task. In addition, we show the parameters learned by SCIPR can be used to align data not used in the training and to identify key cell type-specific genes.

## Author Summary

Integrating single cell expression data (scRNA-Seq) across labs, platforms, and technologies is a major challenge. Current methods for addressing this problem attempt to align cells in one study to match cells in another. While successful, current methods are unable to learn a general alignment in *gene space* that can be used to process new or additional data not used in the learning. Here we show that the scRNA-Seq alignment problem resembles a well known problem in the field of computer vision and robotics: pointcloud registration. We next extend traditional iterative rigid-object alignment methods for scRNA-seq while satisfying a set of unique constraints that distinguishes our solution from past methods. Analysis of transcriptomics data demonstrates that our method can accurately align scRNA-seq data, can generalize to unseen datasets, and can provide useful insights about genes active in the cells being studied.

This is a *PLOS Computational Biology* Methods paper.

**Data Availability Statement:** All scRNA-seq data files are available from the GEO database (accession numbers GSE118767, GSE84133, GSE132044.).

**Funding:** This work was partially funded by the National Institutes of Health (NIH)

(https://www.nih.gov) [grants 1R01GM122096 and OT2OD026682 to Z.B.J.]. The funders had no role in study design, data collection and analysis, decision to publish, or preparation of the manuscript.

## Introduction

While only recently introduced, single-cell RNA-sequencing (scRNA-seq) has quickly developed into an indispensable tool for transcriptomics research. Driven by the development of droplet microfluidics-based methods [1–4] and split-pool barcoding-based methods [5, 6], current experiments are able to simultaneously profile expression of genes in tens of thousands of single cells. Studies ranging from cell type and state identification [7, 8] to tracking early development [9, 10] to unveiling the spatial organization of cells [11, 12] are all utilizing scRNA-Seq data, providing new insights about the activity of genes within and between cells.

While the size and number of individual scRNA-seq datasets is large and constantly growing, the question of how to integrate scRNA-Seq data from multiple experiments or platforms has become increasingly relevant. Different labs are seeking to analyze related tissues in an organ system, such as mapping out the cell types in the human pancreas [13] or building an adult mouse brain cell atlas [14]. On on even larger scale, consortia such as the Human Cell Atlas [15, 16] or the HUBMaP [17] are organizing researchers globally with the goal of mapping cells in the entire human body.

Combining datasets, even for the same tissue, across platforms or labs is a challenging problem. This process is often referred to "dataset alignment", "dataset harmonization", or "batch correction," and is an active area of research. A number of methods have been recently suggested to address this problem. Many of these rely on nearest neighbors computations. For example, Mutual Nearest Neighbors (MNN) integrates two datasets by first identifying cells in the two datasets that are mutual nearest neighbors (in each other's set of $k$ nearest neighbors) [18]. It then computes vector differences between these pairs and uses weighted averages of these vector differences to shift one batch onto the other. Another method, Seurat [19], extends this idea by first computing MNNs in a reduced dimension space, via canonical correlation analysis (CCA) which identifies common sources of variation between the two datasets, and then proceeding to correct the batch effects in a similar fashion as MNN. Other methods such as scVI [20] and ScAlign [21] use a neural network embedding to align the two datasets. These methods seek to encode the scRNA-seq datasets using a common reduced dimensional space in which the batch effects are reduced. While the above methods are unsupervised, there are also a few supervised methods proposed for this task. These method require as input the correct cell type labels for cells in the training data and use that to learn a function to assign cell types for the test data. An example of such method is Single Cell Domain Generalization Network (scDGN) which uses a supervised neural network trained with adversarial objectives to improve cell type classification [22]. Another example is Moana, which uses hierarchical cell type classifiers robust to batch effects to project labels from one dataset onto another [23].

Each of the methods mentioned above offers different features and so might be appropriate for different settings. For example, some methods align the data in the given gene space and thus maintain gene semantics while others, namely the neural network-based methods, do the alignment in a new embedded space (i.e. a reduced dimensional space). On the other hand, the neural network methods typically are learning an alignment function which enables the alignment to be applied to new data (generalization). A comparison of the features of several popular methods is summarized in Table 1. See also [24–26] for recent reviews comparing different alignment methods.

As the table shows, none of the current methods enable both maintenance of semantics (required for analyzing genes following the alignment) and generalization (required for keeping the alignment consistent when new data arrives). Here we propose a new method, Single Cell Iterative Point set Registration (SCIPR), which achieves both using an unsupervised framework. Our method extends a well known method in image analysis termed iterative

**Table 1. Comparison of features and properties of various scRNA-seq alignment methods.**

| Method | Unsupervised? | Corrects input? | Maintains semantics? | Generalizable? | Transfers labels? |
|---|---|---|---|---|---|
| scDGN | | X | | X | X |
| Moana | X | | | | X |
| ScAlign | X | X | | X* | |
| scVI | X | X | | X | |
| Harmony | X | X | | | |
| Scanorama | X | X | X | | |
| MNN | X | X | X | | |
| Seurat | X | X | X | | |
| *SCIPR* | X | X | X | X | |

The "Corrects input?" column refers to whether the method actually aligns (transforms) the input data batches in order to integrate them. The "Maintains semantics?" column refers to whether the output of the method retains the gene semantics given as input. The "Generalizable?" column refers to whether the method learns a model which can be applied to new data. The "Transfers labels?" column refers to whether the method also explicitly aims to apply the cell type labels of one data batch onto another, unlabeled batch.

* ScAlign is theoretically able to be applied on new data, as it learns a neural network embedding model, but the ability to save and load the function in different sessions to apply it on new data was not available in software at the time of testing.

closest points (ICP), which is used for the problem of point-set or point-cloud registration [27]. In ICP, two datasets are represented as sets of points in a common coordinate system, and the method proceeds by pairing together points between the two sets and learning a transformation to move one set closer to (the corresponding points) in the other [28]. A method based on ICP can maintain semantics because it operates in the input data space, meaning that if the input features are genes, then the output features will also be those same genes. Such methods can also generalize to unseen data because they are fitting a transformation function, which can then be applied to new batches.

We tested SCIPR on three benchmark datasets and compared its performance to several prior methods suggested for the alignment task. As we show, single cell iterative point set registration outperforms prior methods for most of the tasks and is able to generalize to both unseen data in the target and in the source batch by learning a general function which can be applied to new data. Finally, since it retains the original (gene space) representation, the coefficients learned by single cell iterative point set registration can be used to identify key genes related to the cell types being analyzed.

## Materials and methods

### Dataset selection

To evaluate SCIPR and to compare its performance to previous alignment methods we used different scRNA-Seq datasets, each profiling similar cells in multiple batches. The first is the CellBench dataset (GEO: GSE118767) [29], which profiled human lung cancer cell lines and contained three batches, each from a different platform: 10x Chromium [4], Dropseq [2], and CEL-seq2 [30] (S1(a) Table). The smallest batch had 210 cells (Dropseq) and the largest had 895 (10x Chromium) after removing cells with low reads, and we filtered the genes to the most highly variable genes across all batches leaving us with 2351 genes (S1 Appendix section "Data preprocessing and filtration"). The second was data from human pancreatic cells (GEO: GSE84133) [31], with four batches all using Indrop sequencing [3] where the largest batch had

1488 cells (inDrop3), the smallest had 834 cells (inDrop4), and we used the five largest cell types and the set of 2629 highly variable genes (S1(b) Table). Finally, the third and largest dataset is a PBMC dataset (GEO: GSE132044) [32] which consisted of four different batches using 10x Chromium [4] sequencing. We used the three largest cell types and the largest batch had 2510 cells (10x Chorm. (v2)), the smallest had 2011 cells (10x Chrom. (v2) A), and we used the set of 1466 highly variable genes (S1(c) Table). See S1 Appendix section "Data preprocessing and filtration" for complete details, and S1 Table for exact numbers of cells in each batch and their cell type distributions.

## scRNA-seq alignment

In the scRNA-seq alignment task, our goal is to learn a new representation of the data (either in the same dimensions as the original data, or in a new reduced dimension) to accomplish the following:

**Property 1** *Cell type identification: Cells from different cell types are distinct and cells from the same type are in close proximity*

**Property 2** *Batch mixing: Cells from different batches are mixed together as much as possible while respecting the first property*

## Point set registration for single cell alignment

Unsupervised alignment of single cell data relies on the implicit assumption that the different datasets share several of the same cell types though potentially using different representations for the same type. A similar assumption is central to much of the literature in point set registration, a well-studied problem in the robotics and computer vision fields [27]. In the point set registration problem, we wish to assign correspondences between two sets of points (two "point clouds"), and learn a transformation that maps one set onto the other (Fig 1). Point sets are commonly the 2D or 3D coordinates of rigid objects, and the class of transformation function under consideration is often rigid transforms (rotations, reflections, and translations). The various point sets often originate from differing settings of sensors (viewing angle, lighting, resolution, etc) viewing the same objects or scene. Among the most widely used and classical of point-cloud registration algorithms is Bessl and McKay's Iterative Closest Point (ICP) algorithm [28]. Briefly, each iteration of ICP has two steps: 1) assigning each point in one set ($A$, "source") to its closest point in the other set ($B$, "target"), 2) update the rigid transformation function to transform the points in $A$ as close as possible to their assigned points in set $B$. At the end of each iteration, the points in $A$ are transformed via the current rigid transform and the process is repeated until convergence. Thus, each iteration of ICP can be concisely represented as minimizing the following loss function:

$$L_{ICP}(A, B, f_\theta) = \sum_{i \in A} \min_{j \in B} \frac{1}{d} ||f_\theta(A_i) - B_j||_2^2 \tag{1}$$

where $A, B \in \mathbb{R}^d$, and $d$ is the number of genes and $f_\theta$ is further constrained to rigid transformation functions

However, applying ICP as-is to align two scRNA-seq datasets could be problematic since:

- ICP assumes that every point in $A$ corresponds to a point in set $B$, whereas scRNA-seq datasets may not fully overlap in cell types. For example, in studying embryonic development, we observe the transcriptome at different embryonic days, where some cell fates exist only after a certain day [33, 34].

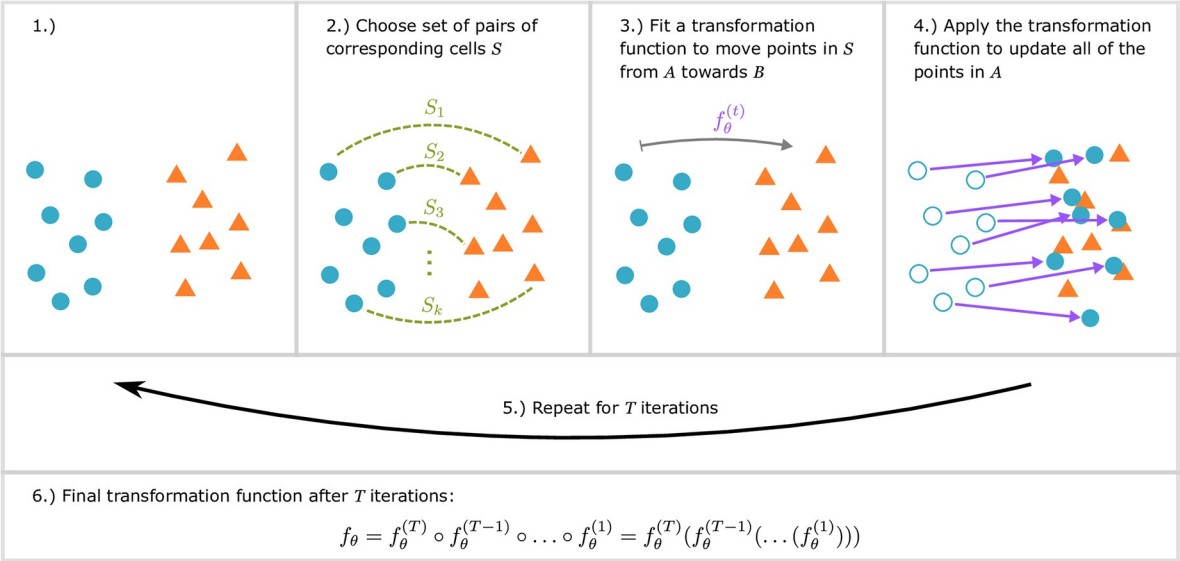

**Fig 1. Summary of steps in iterative point set registration for scRNA-seq data.** Each cell in an scRNA-seq dataset can be viewed as a point in high dimensional space. 1) We start with two unaligned batches (sources, blue and targets, orange). 2) A matching algorithm (e.g. picking the closest corresponding point, or using mutual nearest neighbors) is used to pair source cells from $A$ with a corresponding target cell in $B$. The number of source and/or target cells matched can vary for different matching strategies. 3) Based on the selected pairs, a global transformation function is learned so that source cells in $A$ become closer to their paired cell in $B$. 4) The learned transformation is next applied to all points in $A$. 5) This process (steps 2–4) is repeated, iteratively aligning set $A$ onto $B$ until the mean distance between the assigned pairs of cells no longer improves. 6) The final global transformation function is the composition of the functions learned in each iteration at step 3.

- ICP assumes that a rigid transform relates the two sets. This may have been appropriate for 3D rigid objects, but not for the complicated, high-dimensional single cell transcriptome data.

- ICP is prone to assigning many points in $A$ to the same point in $B$ (collapsing a point) even when they are not fully compatible [35]. In contrast, if the same cell type exists in both datasets we can expect the number of cells to be more balanced.

Thus, while ICP has been very successful in image analysis, it requires modifications in order to accurately align scRNA-Seq data.

### Adapting ICP for scRNA-seq dataset alignment

Given the discussion above, both stages of ICP need to be changed in order to align scRNA-Seq data. More formally, these two stages are:

1. Assignment stage (input: point sets $A$ and $B$)—assign pair set $S \subseteq \{(i, j) \mid 1 \le i \le |A|, 1 \le j \le |B|\}$ (Fig 1 panel 2). Options for this stage may vary in the cardinality of $S$, and whether or not it allows points to be shared between pairs.

2. Transformation stage (input: assigned pairs $S$)—given $S$ from the Assignment stage, learn a transform function that transforms points in $A$ to reduce the mean squared error (MSE) between the assigned pairs in $S$ (Fig 1 panel 3). Options for this stage vary based on the family of functions considered.

To adapt stage 1, we propose two approaches for assigning points in the next section: one based on a novel greedy algorithm and another based on Mutual Nearest Neighbors (MNNs). For stage 2, we set the family of transformation functions to be affine transformations (section titled "Learning a transformation function").

## Assigning cells between datasets

First, we focus on the Assignment stage. The input to this stage is the target set $B$ and the current state of the source set of points $A$ ($A$ is being updated at every iteration of the algorithm). ICP computes the pairwise distance matrix between members of these sets $\boldsymbol{D} \in \mathbb{R}^{n \times m}$ where $n = |A|$ and $m = |B|$ (S1 Fig), and finds the element in each row with the smallest distance to match to that point in $A$.

In contrast, for scRNA-seq alignment we would like to require the following:

- Not too many points in $A$ are matched with the same point in $B$ (avoid collapsing many points in one dataset onto a single point).

- Not all points must be assigned (since the two dataset may not fully overlap in terms of cell types).

One approach for addressing the first requirement is using a bipartite matching algorithm [36] instead of picking the closest point. In such an algorithm a global optimal matching is found such that each point is only matched to a single point in the other set. However, such algorithms violate the second requirement since they result in "perfect" matchings, where all points in $A$ are matched. An alternative is to use partial matching algorithms in which only a subset (or a fraction) of the points in A are required to be matched to points in B. Optimal partial matching is a well studied problem in the computer science literature and requires solving a min-cost flow graph problem [37]. The problem can be solved via an efficient network simplex algorithm, however, for graphs with thousands of nodes (as in single cell data) this is still rather time consuming. If we let the number of vertices be $V = n + m$ (e.g. number of cells in both batches), the number of edges be $E = n \times m$, and the largest edge weight (distance between points) be $C$, then the polynomial time network simplex algorithm has a run time of $O(VE \log V \log(VC))$ [38]. Given the large number of cells in each dataset such partial matching methods are too time consuming in practice (S2 Fig).

Instead, in S1 Algorithm we propose an efficient greedy algorithm for partial assignment between $A$ and $B$. The algorithm sorts all of the edges between members of the two sets based on distance. Next, we proceed along the ordered edges starting from the smallest distance. If an edge includes a point in set $B$ that has already been selected $\beta$ times by previously chosen edges, we discard it and continues down the list. Our algorithm has parameters to adjust how many times we allow each point in $B$ to be matched to ($\beta$), and how many of the points in $A$ must be matched ($\alpha$). In our experiments, we set $\beta = 2$ to allow more flexibility than bipartite single-matching, while strictly preventing over-matching and the collapse of several cells onto the same point and $\alpha = 0.5$ because requiring 50% matching would allow for cases where significant portions of the source or target cell types do not overlap. These hyperparameters can be more precisely set based on the user's prior belief of the composition of their batches, though we show that these default settings worked well in our experiments across three datasets. The runtime of this algorithm is dominated by the `sortElementsAscending` function which sorts the distances leading to a worst case runtime of $O(E^2)$ and a much faster $O(E \log(E))$ on average. Though not an optimal solution to the partial bipartite matching problem, we find that this works well in our related scRNA-seq cell pair assignment problem for alignment.

We also experiment with a matching procedure which follows the foundational work of using mutual nearest neighbors (MNNs) [18] to define our pair assignments between the two sets. For a point $i$ in set $A$ and a point $j$ in set $B$, if $i$ is in the set of $k$-nearest neighbors among $A$ for point $j$, and $j$ is in the set of $k$-nearest neighbors among $B$ for point $i$, then $i$ and $j$ are MNNs. In our experiments, we set $k = 10$. To pick this default value for $k$, we used the average of the defaults in two well-established analysis frameworks for scRNA-seq, Seurat [39] (`FindIntegrationAnchors` function which uses $k = 5$) and scanpy [40] (`scanpy.pp.neighbors` which uses $k = 15$) and we found that such assignment worked well in our experiments across all three datasets.

## Learning a transformation function

So far we focused on matching points given their distance. In the next stage, we will fit a transformation function to align the matched points. As discussed above, the family of rigid transforms is not well suited to compute such alignments for scRNA-seq data (S3 Fig). Instead, we propose to use the family of affine transformations to align scRNA-seq datasets. Affine transforms are of the form $f_\theta(x) = W^T x + b$ where $\theta = \{W, b\}$ is the learnable weights of the function, and include rotation, reflection, scaling, and shearing.

To learn this function, we minimize an objective function that aims to move the assigned points closer to each other, via such an affine transformation. Given a pair assignment $S$ from the previous step (section "Assigning cells between datasets") (which may be the result of the classic "closest" strategy from ICP, our greedy algorithm, or MNN matching), learning the transformation function (Fig 1 panel 3) is equivalent to minimizing the loss function given as Eq (2). We note that this objective function is not over all pairs of points in sets $A$ and $B$; it is computed over only those pairs of points selected in $S$, denoted by the subscript under the sum.

$$
\begin{aligned}
L(A, B, f_\theta, S) \quad &= \frac{1}{|S|} \sum_{i,j \in S} \frac{1}{d} \left|\left| f_\theta(A_i) - B_j \right|\right|_2^2 \\
&= \frac{1}{|S|} \sum_{i,j \in S} \frac{1}{d} \left|\left| (W^T A_i + b) - B_j \right|\right|_2^2
\end{aligned}
\tag{2}
$$

where $A, B \in \mathbb{R}^d$, and $d$ is the number of genes.

This is a least-squares objective function. If the system is overdetermined, this could be solved exactly. However, due to the high dimension we are working in (each point is the expression of thousands of genes), the matrix inversion for the exact solution is expensive to compute, as matrix inversion is $O(d^3)$. To avoid this, we approximate the solution using gradient descent to arrive at our transformation function $f_\theta^{(t)}$ for the current iteration $t$ (see S1 Appendix section "Parameter settings for SCIPR experiments" for gradient descent settings).

## Iterative step

After each of the stages (assignment and transformation function update), we use our learned transformation function at the current iteration $f_\theta^{(t)}$ to transform the all points in $A$ (not just those in the set $S$ from the matching algorithm) (Fig 1 panel 4), and then repeat the stages for $T$ iterations (Fig 1 panel 5). The final learned transformation of source points $A$ to target points $B$ is a chained series of transformations (composite function) from each iteration (Fig 1 panel 6). Since our function class for $f_\theta$ is affine transformations, and the composition of affine transformations is itself an affine transformation, we can combine this chain of transformations

into a single affine transformation. See S1 Appendix section "Computing the final affine transformation at the end of SCIPR" for details.

In all of our experiments, we ran the iterative point set registration for five iterations. Our experiments (S9 Fig) indicate that distances between matched cells converge within very few iterations for all three datasets. We note that this is in line with prior work that used the ICP algorithm on image data. For that data ICP also exhibited fast convergence within the first few iterations [28].

The runtime of our iterative algorithm is slower than that of the pure matching-based methods such as MNN and Seurat (S6 Table), and is similar to the runtime of neural network-based methods such as ScAlign. This is because while MNN and Seurat do not learn a function and must be recomputed to align any new data, our point set registration method and methods like ScAlign aim to learn an alignment function that can generalize and be applied to align new data not seen in the learning process. This comes at the cost of having to optimize an objective function via an iterative learning procedure. However, we note that these methods can utilize graphics processing units (GPUs) to greatly accelerate the process, and we see that our iterative algorithms can be even faster than MNN or Seurat when run with a GPU (S6 Table).

## Validation

A number of methods have been proposed to test the accuracy of alignment based methods [41, 42]. These evaluation metrics try to balance two, sometimes competing, attributes. The first is dataset mixing which is the goal of the alignment. The second is cell type coherence. A method that randomly mixes the two datasets would score high on the first measure and low on the second while a method that clusters each of the datasets very well but cannot match them will score high on the second and not on the first.

To track both dataset mixing and biological signal preservation, we follow [42] and use the local inverse Simpson's Index (LISI). LISI measures the amount of diversity within a small neighborhood around each point in a dataset, with respect to a particular label. The lowest value of LISI is 1 (no diversity). As in [42], we define integration LISI (iLISI) as the score computed when using the batch label for each datapoint, and cell-type LISI (cLISI) as the score when using the cell-type label. iLISI measures the effective number of datasets within the neighborhood (so the higher the better). cLISI measures the effective number cell types within the neighborhood (so the lower the better). See [42] and S1 Appendix section "Computing iLISI and cLISI scores" for details on how to compute LISI scores (Equation S1). With these two metrics in hand, we can keep track of not only the ability of our algorithms to align one dataset onto another, but also their ability to preserve original signal. In our figures which report the iLISI and cLISI scores, we rank the methods based on the difference of medians $iLISI - cLISI$ score to capture the ability of the methods to maximize and minimize these two quantities respectively.

## Results

### Method and benchmarking overview

We developed SCIPR which aligns two batches of scRNA-seq data (termed source and target) using methods motivated by point set registration algorithms. SCIPR first identifies corresponding pairs of cells between source and target batches (Fig 1 panel 1). Rather than using the closest cell (as defined by euclidean distance) in the target to match a source cell, SCIPR uses either of two matching algorithms to account for the heterogeneity and noise in scRNA-seq data: Mutual Nearest Neighbors (MNN) matching [18], and a novel greedy matching

algorithm (S1 Algorithm, Methods). Once a pairing of cells is established (Fig 1 panel 2), a transformation function is learned to transform source cells so that they are closer to their matched target cell (Fig 1 panel 3). To allow for accurate alignment of high-dimensional scRNA-seq data, we replace the rigid transformation commonly used for point cloud registration with affine transformations. After fitting the transformation function (Methods), we apply it to the source cells (Fig 1 panel 4), and iteratively repeat the process until convergence. The final alignment function we learn is a composition of the transformation functions learned at each iteration (Methods) (Fig 1 panels 5 and 6).

We used three datasets to test and compare two versions of SCIPR to prior alignment methods (Methods). See S1 Appendix section "scRNA-seq alignment benchmarking software and data" for software settings of the related methods. These comparisons were performed by testing the methods on several "alignment tasks". An alignment task is defined by:

- A dataset (e.g. Pancreas)

- A source batch *A* within that dataset, which you would like to transform (e.g. inDrop1)

- A target batch *B* within that dataset, which you would like to transform *A* onto (e.g. inDrop 3)

For example, an alignment task can be summarized with the notation: *Pancreas: inDrop1 → inDrop3*. In the comparisons we performed we fix the target within a dataset to be the largest batch in that dataset. We define these tasks as pairwise alignments, but we note that it is possible to use our method to align multiple batches using SCIPR (S11 Fig). We scored the performance of the methods using local inverse Simpson's Index (LISI) in which higher integration LISI (iLISI) is better and lower cell-type LISI (cLISI) is better [42] (Methods).

## An affine global transformation function yields well-mixed alignments

We first evaluated the ability of SCIPR and other methods to integrate pairs of batches from three different datasets. Results for 8 alignment tasks in three datasets are presented in Fig 2. As the figure shows, for 7 of the 8 alignment tasks the two version of SCIPR ranked at the top. SCIPR-mnn was the overall top performer ranking first on 4 tasks and 2nd on 2 whereas SCIPR-gdy ranked first on 3 tasks and 2nd on 1. The only other method that performed well is ScAlign which ranked first on 1 task and 2nd on 4. For example, for the CellBench alignment tasks (first row of Fig 2), we see that SCIPR-mnn, which uses the MNN matching for the cell pair assignment stage, has consistent better performance, and achieves high batch mixing (1.70 and 1.76 median iLISI scores on *CELseq2→10x* and *Dropseq→10x* respectively) with very little cell type mixing (1.00 median iLISI score on both *CELseq2→10x* and *Dropseq→10x*). When looking at the same dataset, on the *CELseq2→10x* task the other methods such as ScAlign (iLISI: 1.00, cLISI: 1.00) or SeuratV3 (iLISI: 1.51, cLISI: 1.00) are also able to avoid cell type mixing, but are not able to mix the batches as much as SCIPR (Fig 2). Full alignment quantitative scores for these tasks and all others in the paper are listed in S7 Table. These quantitative metrics are also corroborated by a qualitative assessment of the resulting t-SNE embeddings (Fig 3). There we can see that both SCIPR-gdy and SCIPR-mnn (top two rows) mix the batches well (1st and 3rd columns) compared to methods like MNN and ScAlign while successfully keeping cell types separate (2nd and 4th columns). SeuratV3 also performs well. The embeddings on the Pancreas dataset (S10 Fig) show that most methods result in embeddings with these desirable properties on this dataset, including SCIPR.

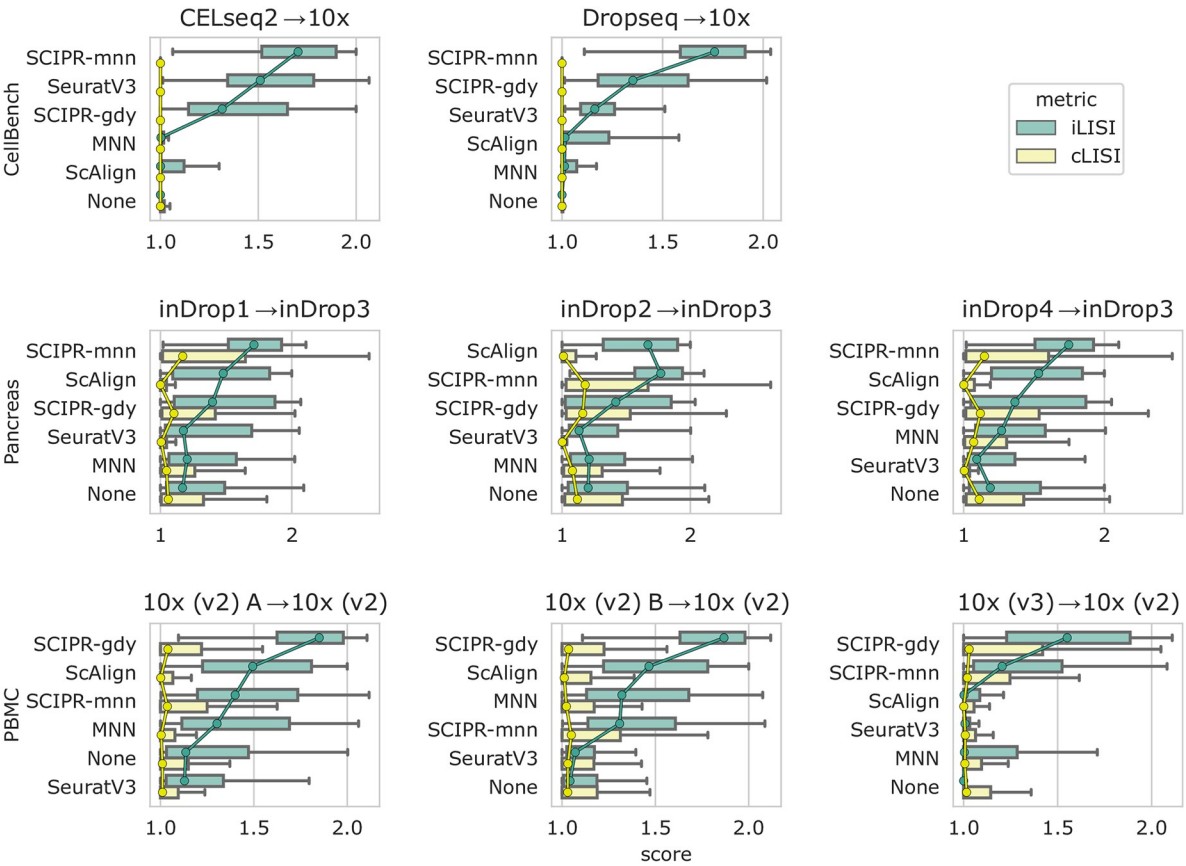

**Fig 2. Quantitative scoring of alignment methods on benchmark datasets.** Each row of subplots are tasks from the same dataset, where each column uses a different source batch (all are aligned to the same largest reference batch). The scores are iLISI (green, batch integration score), and cLISI (yellow, cell type mixing score). In each subplot the methods are ordered from top to bottom in order of largest difference (median iLISI – median cLISI) of scores. "None" means no alignment method is applied to the data. The center of each box is the median, and whiskers represent 1.5 times the IQR past the low and high quartiles. Circle markers are placed on the medians and connected between boxes with lines of the corresponding color to facilitate visual comparisons.

## SCIPR robustly mixes batches with non-overlapping cell types

The comparisons presented above involved sources and target batches with the same set of cells. However, in practice it is often unknown if both source and target indeed contain the same cell types. To test the robustness of SCIPR and other methods for such realistic scenarios we hold-out a complete cell type from the target set *B* in each of the alignment tasks from the section "An affine global transformation function yields well-mixed alignments". As the figures show, for these alignment tasks SCIPR is able to mix batches well, while keeping the median cell type mixing (iLISI) score low, though with a longer tail (Fig 4 and S4, S5 Figs). For example, for the *CellBench: CELseq2→10x (H1975 cell type held-out from target)* task, SCIPR-mnn had median iLISI and cLISI scores of 1.63 and 1.02 respectively while the second best method, SeuratV3, had iLIS and cLISI scores of 1.49 and 1.01 respectively (Fig 4). On the other hand, for the task *Pancreas: inDrop1→inDrop3 (acinar cell type held-out from target)*, SCIPR-mnn achieves a higher median batch mixing score of iLISI = 1.69 compared to ScAlign's score of 1.57, but also mixes the cell types slightly more (SCIPR-mnn median cLISI score: 1.28, ScAlign median cLISI score:1.00) (S4 Fig).

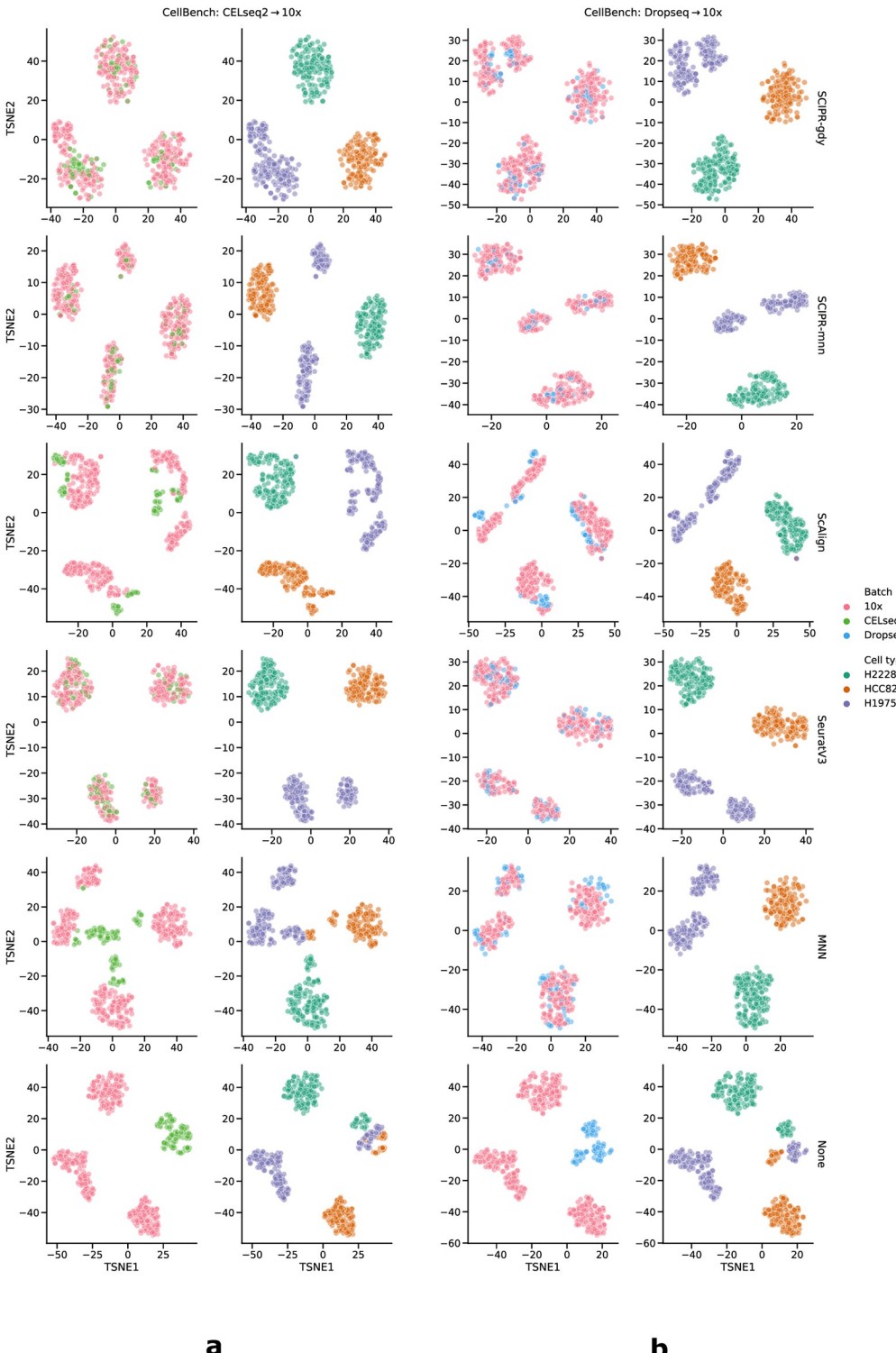

**Fig 3. Embedding (t-SNE) visualization from alignment tasks on the CellBench dataset using various alignment methods.** Each row is a different alignment method (the bottom row, "None", is with no alignment). The columns are in two groups based on alignment task: the left two columns (a) pertain to aligning the CELseq2 batch onto the 10x batch, the right two columns (b) are for aligning the Dropseq batch onto the 10x batch. The first and third columns are colored by batch, and the second and fourth columns are colored by cell type.

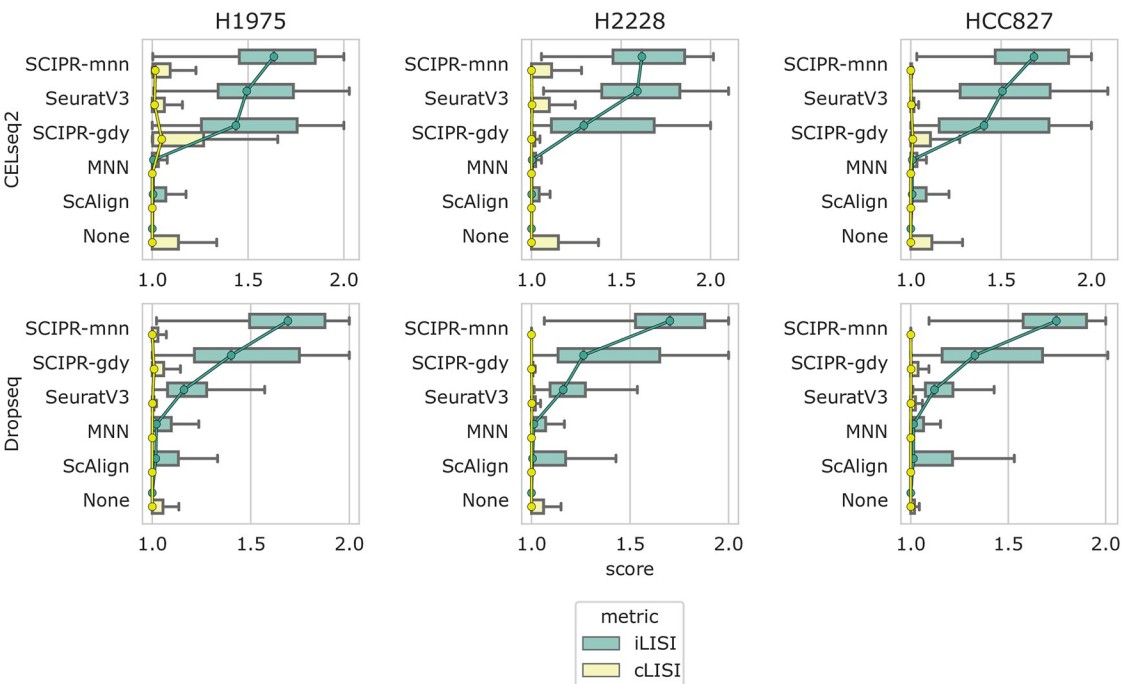

**Fig 4. Quantitative scoring of alignment methods on the CellBench dataset with a cell type held out from the target set.** Each row of subplots are alignment tasks with the same source batch, where each column uses a different cell type as a hold-out from the target set (the 10x batch). Box plot computation and ordering of methods in each subplot is determined in the same fashion as in Fig 2

## SCIPR generalizes to unseen data

One of the advantages of SCIPR compared to most previous methods is the fact that it learns a general transformation function that can be applied to additional data when it becomes available (Table 1). Such a function allows researchers to "fix" a specific setting rather than have all results completely change when new data is introduced. To test the use of the learned transformation function for unseen cell types in the *source dataset* we repeated our analysis, this time holding out a complete cell type from the source set in each alignment task. We next learned the transformation based on the available data and then applied the learned function to the held out data to evaluate the batch and cell type mixing. Results are presented in Fig 5, S6, S7 and S8 Figs. As the figures show, the transformation learned by SCIPR allows it to keep cell types distinct, even for the unseen source cell type, while also being able to mix the batches of unseen cell types. This is evident in the high median iLISI (1.69) and low median cLISI (1.04) scores of SCIPR-gdy on the task *PBMC:10x Chrom. (v2) A→10x Chrom. (v2) (CD4+ T cell held-out from source)*, where the model is fit without seeing CD4+ T cells in the source set, but is then used to transform the full source set in evaluation (S8 Fig). Fig 5 displays the aligned results for the cell type not used in the learning. As the figure shows, for CD4+ and Cytotoxic T cells SCIPR-gdy is able to mix the two batches even though it had never seen these in fitting.

## SCIPR identifies biologically relevant genes

The above results demonstrate SCIPR's ability to integrate batches quantitatively and qualitatively. Since SCIPR achieves these results by learning a transformation function that places different weights on different genes, we next asked whether the learned weights provide

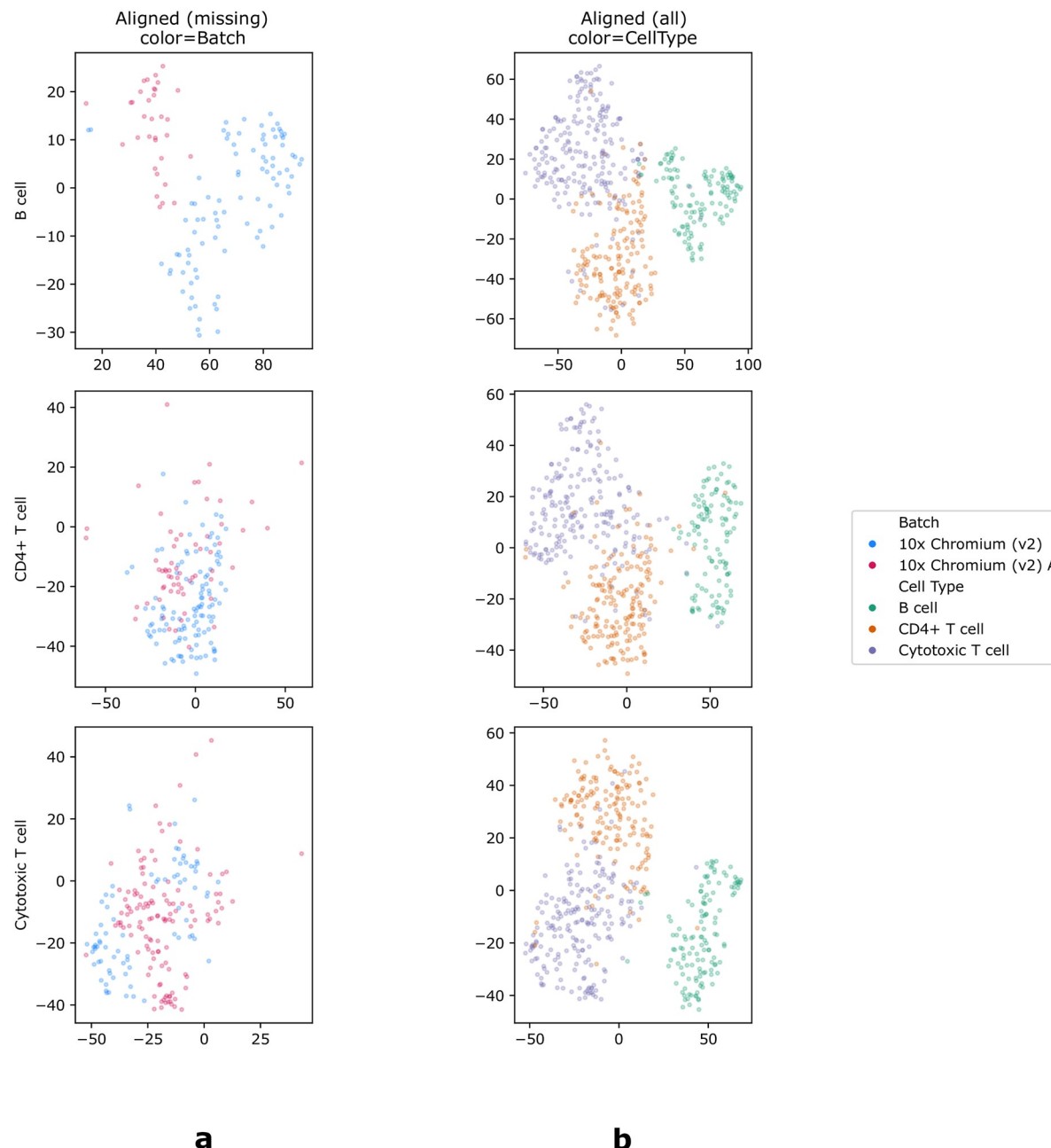

**Fig 5. Embedding (t-SNE) visualization from the *PBMC:10x Chrom. (v2) A→10x Chrom. (v2)* task using SCIPR-gdy showing generalizability to new cells.** In each alignment task (rows), a different cell type is completely held-out from the *source* set. The model is then fitted to align the source and the target, and this fitted model is then used to transform the full source set, including the held-out cell type which the model did not see in the source set used for fitting. The first column (a) shows just the held-out cell type colored by batch, after applying the fitted SCIPR-gdy model to align it. The second column (b) shows all of the data after applying the fitted model, colored by cell type.

information on the importance of specific genes for the set of cells being studied. Since SCIPR aims to align specific pairs of cells (one from each batch), when it is successful it tends to focus more strongly on cell type-specific genes. As we showed, for several datasets the method is indeed correct in the assignments it identifies and for these, the set of genes it uses may be of

relevance for the cell types it aligns (Sections "An affine global transformation function yields well-mixed alignments" and "SCIPR robustly mixes batches with non-overlapping cell types"). To evaluate this idea we compared ranking genes based on their SCIPR coefficients to a baseline that ranks them based on differential expression (DE) (S1 Appendix section "Differential expression analysis" and S1 Appendix section "Selecting top genes from SCIPR models for enrichment analysis"). Next we performed gene set enrichment analysis using the Gene Ontology to identify the significant functions associated with top genes and test their relevance (S1 Appendix section "Gene set enrichment analysis"). The PBMC dataset, which is the largest, was also the one with the most number of significant categories identified (S2 Table). When comparing top ranked genes by SCIPR and DE for the PBMC: *10x Chrom.* (*v2*) *A  10x Chrom.* (*v2*) alignment task we observed that SCIPR genes significantly overlapped with much more relevant terms when compared to DE genes for the same dataset (S3 and S4 Tables for SCIPR-mnn and SCIPR-gdy respectively). For example, the top three categories for top ranked SCIPR-mnn genes are "Defense response", "Regulation of immune response", and "Humoral immune response" (all with adj. p-value 9.743e-9, S3 Table). These categories are very relevant for blood cells given their immune system function. On the other hand, the top three categories recovered by top DE genes are much more generic and include "Cotranslational protein targeting to membrane", "Cellular amide metabolic process", and "Organic cyclic compound catabolic process" (S2 Table). We compared the enrichment results obtained from using top ranked genes based on SCIPR to results obtained using highly variable genes (HVG). While HVG's do result in enrichment of relevant GO terms (S5 Table), the significance of enriched relevant categories is lower when compared to genes ranked by SCIPR-mnn.

## Discussion

We presented SCIPR which extends point set registration for the alignment of scRNA-Seq data. SCIPR combines many of the desirable features of previous methods including the fact that its unsupervised, generalizable, and keeps the original (gene space) representation. Analysis of several datasets show that SCIPR successfully aligns scRNA-Seq data improving upon other methods proposed for this task. When data is missing from either the source or the target the transformation function learned by SCIPR can be used to accurately align it when it becomes available. Finally, the coefficients learned by SCIPR provide valuable information on the key genes related to the cells being analyzed.

Framing scRNA-seq alignment as a point set registration problem opens the door to applying many of the developments and advancements in that area to scRNA-seq alignment. Point set registration is a mature area that has been widely used for more than two decades. As part of this researchers looked at several different types of transformation functions, data filtration, outlier handling, and association mapping, all of which may find applications in scRNA-seq analysis.

When evaluating SCIPR and prior methods we used the local inverse Simpson's Index (LISI) to quantify both cell type mixing and batch mixing. This leads to two values for each alignment task which can be combined for ranking the different methods by computing the difference of the medians $iLISI - cLISI$. Such ranking places equal weight on both issues. However, this score may not tell the whole story since some methods may be much better at one task vs. the other. For example, while SCIPR was ranked as the top method for most of the comparisons we performed, it has a tendency of to sacrifice some cell type separation in order to achieve greater batch mixing. Thus, depending on the user priorities between cell type and batch mixing, different methods may be more attractive even if the combined score is lower when compared to other methods.

With regards to choosing between SCIPR-mnn and SCIPR-gdy, we generally recommend the use of the MNN strategy which performs well and does not rely on hyper-parameters related to the expected matching between the datasets (thus, requires less assumptions). However, in cases where the user has prior beliefs or expectations on what portion of the cells should be matched between the two batches (for example, if s/he knows that some types are missing in one sample and so the match proportion is not expected to be high) then we recommend using the greedy algorithm, since such information can be incorporated using its hyper-parameters and will thus lead to better results. We tested SCIPR using a set of highly variable genes (see "Dataset selection" in Methods and S1 Appendix section "Data preprocessing and filtration"). Previous methods also recommend using highly variable genes [19]. Using such set limits the application of our alignment transformation to a rather small subset of genes, and so may have implications for downstream analysis. We note that SCIPR can be run with more genes, though we observe a modest decrease in performance when non-variable genes are included (S12 Fig), only showing a slight decrease in alignment performance in some cases. While the set of highly variable genes alone does result in enrichment for relevant terms (S5 Table), the significance of this enrichment is not as strong as the results from our SCIPR-mnn model weights analysis.

While SCIPR performed best in our analysis, there are a number of ways in which it can be further improved. As mentioned above, SCIPR tends to weight batch mixing higher than cell type separation. A possible way to overcome this would be to add a regularization term to the transformation function to increase the weight of high scoring matches. Another option is to explore the use of non-linear transformations with strong customized regularization. A potential drawback of SCIPR is the fact that it relies on affine transformations that do not handle non-linear differences between batches. While this leads to a reduction in the number of parameters that the method uses, and so reduces overfitting, it also means that the method cannot handle non-linear changes which may occur between different batches.

SCIPR is implemented as a Python package, with documentation, installation instructions, and source code available at https://scipr.readthedocs.io, and our benchmarking pipeline and data used are available at https://github.com/AmirAlavi/sc-alignment-benchmarking (see S1 Appendix section "scRNA-seq alignment benchmarking software and data").

## Supporting information

**S1 Fig. The distance matrix between cells in batches *A* and *B*.** $D \in \mathbb{R}^{n \times m}$ where $n = |A|$ and $m = |B|$ and $D_{i,j}$ is the distance between cells $A_i$ and $B_j$. In our work, $A$ is called the "source" set and $B$ is called the "target" set. We use the euclidean distance throughout our paper. (PDF)

**S2 Fig. A comparison of runtimes of a greedy matching algorithm (S1 Algorithm) compared to a network flow-based approach for finding an optimal partial matching (Min Cost Flow).** For both algorithms, we generated random distances matrices with varying numbers of cells (also called nodes). In this simple case designed to test algorithm runtimes as a funciton of input size, the distances were integers uniformly drawn from [0, 50]. The distance matrices are square, representing the case when a source batch has the same number of cells as the output batch, where the x-axis in the above plots is the number of cells in each batch. For the greedy algorithm, we used the default parameters as discussed in the main body and in S1 Appendix section "Parameter settings for SCIPR experiments". For the Min Cost Flow algorithm, we started by constructing a bipartite graph where nodes in one set represented cells in the source batch, and nodes in the other set represented cells in the target set. The directed (from source to target) edge weights (costs) were set to the distances between the nodes as

given by the randomly generate distance matrix. Then a "source" node was added and connected to all of the nodes of the source cells, and a "sink" node was added and all of the nodes of the target cells were connected to it. The demand of the source node was set to -0.5 × nodes (the number of "units" of flow that this node wants to send, i.e. the number of pairings of cells we want to assign), and the demand of the sink node was set to negative of this (the number of "units" of flow that it wants to receive). Finally, the capacity of each edge in this directed network was set to 1, and the the network simplex algorithm was used to find a solution. The directed graph was constructed using the NetworkX python package and they network simplex algorithm was run via the `min_cost_flow` function [43].
(PDF)

**S3 Fig. Comparison of using a rigid transformation versus an affine transformation in the SCIPR method.** These alignment tasks are from the CellBench dataset, the smallest and somewhat easiest dataset. The two subplots use different source batches (both are aligned to the same largest reference batch, 10x). The scores are iLISI (green, batch integration score), and cLISI (orange, cell type mixing score). In each subplot the methods are ordered from top to bottom in order of largest difference (median iLISI − median cLISI) of scores. The center of each box is the median, and whiskers represent 1.5 times the IQR past the low and high quartiles. Circle markers are placed on the medians and connected between boxes with lines of the corresponding color to facilitate visual comparisons. We can see that even on this rather small dataset, the rigid transformation functions are not sufficient to integrate the data well (low iLISI scores), and we see a large gap in iLISI compared with the affine transformation functions.
(PDF)

**S4 Fig. Quantitative scoring of alignment methods on the Pancreas dataset with a cell type held out from the target set.** Each row of subplots are alignment tasks with the same source batch, where each column uses a different cell type as a hold-out from the target set (the inDrop3 batch). Box plot computation and ordering of methods in each subplot is determined in the same fashion as in S3 Fig.
(PDF)

**S5 Fig. Quantitative scoring of alignment methods on the PBMC dataset with a cell type held out from the target set.** Each row of subplots are alignment tasks with the same source batch, where each column uses a different cell type as a hold-out from the target set (the 10x Chrom. (v2) batch). Box plot computation and ordering of methods in each subplot is determined in the same fashion as in S3 Fig.
(PDF)

**S6 Fig. Quantitative scoring of alignment methods on the CellBench dataset with a cell type held out from the source set.** Each row of subplots are alignment tasks with the same source batch, where each column uses a different cell type as a hold-out from the source set. The target set is 10x for all. In the first column, "None hidden", no cells were hidden from the source set. Box plot computation and ordering of methods in each subplot is determined in the same fashion as in S3 Fig.
(PDF)

**S7 Fig. Quantitative scoring of alignment methods on the Pancreas dataset with a cell type held out from the source set.** Each row of subplots are alignment tasks with the same source batch, where each column uses a different cell type as a hold-out from the source set. The target set is inDrop3 for all. In the first column, "None hidden", no cells were hidden from the

source set. Box plot computation and ordering of methods in each subplot is determined in the same fashion as in S3 Fig.
(PDF)

**S8 Fig. Quantitative scoring of alignment methods on the PBMC dataset with a cell type held out from the source set.** Each row of subplots are alignment tasks with the same source batch, where each column uses a different cell type as a hold-out from the source set. The target set is 10x (v2) for all. In the first column, "None hidden", no cells were hidden from the source set. Box plot computation and ordering of methods in each subplot is determined in the same fashion as in S3 Fig.
(PDF)

**S9 Fig. Convergence during fitting of SCIPR models.** Each row of subplots are tasks from the same dataset, where each column uses a different source batch (all are aligned to the same largest reference batch, 10x for CellBench, inDrop3 for Pancreas, and 10x v2 for PBMC). The values plotted are the mean distances between the selected pairs of points after each iteration of the algorithm. We can see that both SCIPR-gdy and SCIPR-mnn do indeed converge to a local optimum within the first few iterations. Fast convergence within the first few iterations is expected for Iterative Closest Points-based algorithms [28]. This supports our choice to run the SCIPR methods for 5 iterations in the experiments we present in our work.
(PDF)

**S10 Fig. Embedding (t-SNE) visualization from alignment tasks on the Pancreas dataset using various alignment methods.** Each row is a different alignment method (the bottom row,"None", is with no alignment). The columns are in three groups based on alignment task: the left two columns (a) pertain to aligning the inDrop1 batch onto the inDrop3 batch, the middle two columns (b) are for aligning the inDrop2 batch onto the inDrop3 batch, and the right two columns (c) are for aligning the inDrop4 batch onto the inDrop3 batch. The first, third, and fifth columns are colored by batch, and the second, fourth, and sixth columns are colored by cell type.
(PNG)

**S11 Fig. Example of aligning multiple batches to a reference batch using SCIPR.** To see how SCIPR can be used to align multiple batches to a reference batch, we aligned each of the inDrop1, inDrop2, and inDrop4 batches to the inDrop3 batch (the largest batch) in the Pancreas dataset. These alignments were done independently, as pairwise alignments, and visualized together in the figure. In the subplots on the left, each point (cells) is colored by batch, and on the right they are colored by cell type. This straightforward multiple alignment strategy shows that it is possible to align many different batches to a single reference batch using SCIPR which results in coherent cell type representations while mixing the batches well.
(PDF)

**S12 Fig. Comparison of using highly variable genes versus additional random genes in SCIPR.** We compared using highly variable genes (as described in S1 Appendix "Data preprocessing and filtration"), versus using additional genes, for SCIPR models on the PBMC dataset (the largest dataset). Both our SCIPR-gdy and SCIPR-mnn models were run with either just the highly variable genes ("-hvg" suffix) (there are 1466 in the PBMC dataset) or with the highly variable genes and an equal number of randomly selected other genes ("-hvg+rnd" suffix). The three subplots correspond to the three different alignment tasks (aligning a source batch to a target batch) within the PBMC dataset. These quantitative scores show that SCIPR still performs well, and is robust to the inclusion of even more genes that are not necessarily

the most informative genes. Box plot computation and ordering of methods in each subplot is determined in the same fashion as in S3 Fig.
(PDF)

**S1 Table. Cell type and batch distributions for three scRNA-seq datasets we use for evaluation.** Cell type and batch distributions for three scRNA-seq datasets we use for evaluation. Each row pertains to a batch, each column pertains to a cell type, and each value is the number of cells for each row and column combination. Numbers here are *after* our preprocessing described in S1 Appendix section "Data preprocessing and filtration". The largest batch in each dataset is bolded, which we use as our reference "target" batch in our alignment tasks.
(PDF)

**S2 Table. Gene enrichment analysis of Differential Expression results.**
(PDF)

**S3 Table. Gene enrichment analysis of model weights from SCIPR-mnn.**
(PDF)

**S4 Table. Gene enrichment analysis of model weights from SCIPR-gdy.**
(PDF)

**S5 Table. Gene enrichment analysis of highly variable genes.**
(PDF)

**S6 Table. Runtimes of alignment methods.**
(PDF)

**S7 Table. Quantitative alignment scores from all alignment tasks in our work.**
(XLSX)

**S1 Appendix. Supporting methods.** This appendix contains the details of supporting analysis methods, including how we preprocess and filter data, conduct differential expression analysis, discover top weighted genes in our SCIPR models, and conduct gene set enrichment analysis. It contains additional details on computing LISI scores and computing the final affine transformation from SCIPR models. It also includes details of our software pipeline and settings for related methods.
(PDF)

**S1 Algorithm. Greedy pair assignment algorithm.**
(PDF)

## Acknowledgments

We would like to thank all past and present members of the Bar-Joseph Systems Biology Group for valuable input and feedback regarding SCIPR and evaluation methods.

## Author Contributions

**Conceptualization:** Amir Alavi, Ziv Bar-Joseph.

**Data curation:** Amir Alavi.

**Formal analysis:** Amir Alavi, Ziv Bar-Joseph.

**Funding acquisition:** Ziv Bar-Joseph.

**Investigation:** Amir Alavi, Ziv Bar-Joseph.

**Methodology:** Amir Alavi, Ziv Bar-Joseph.

**Project administration:** Ziv Bar-Joseph.

**Resources:** Ziv Bar-Joseph.

**Software:** Amir Alavi.

**Supervision:** Ziv Bar-Joseph.

**Validation:** Amir Alavi.

**Visualization:** Amir Alavi.

**Writing – original draft:** Amir Alavi, Ziv Bar-Joseph.

**Writing – review & editing:** Amir Alavi, Ziv Bar-Joseph.

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
