## [Decision Letter · Decision Letter 0]

25 Jun 2020

Dear Dr. Bar-Joseph,

Thank you very much for submitting your manuscript "Iterative point set registration for aligning scRNA-seq data" for consideration at PLOS Computational Biology.

As with all papers reviewed by the journal, your manuscript was reviewed by members of the editorial board and by several independent reviewers. In light of the reviews (below this email), we would like to invite the resubmission of a significantly-revised version that takes into account the reviewers' comments.

We cannot make any decision about publication until we have seen the revised manuscript and your response to the reviewers' comments. Your revised manuscript is also likely to be sent to reviewers for further evaluation.

Sincerely,

Shi-Jie Chen

Associate Editor

PLOS Computational Biology

Weixiong Zhang

Deputy Editor

PLOS Computational Biology

Reviewer's Responses to Questions

**Comments to the Authors:**

Reviewer #1: Summary:

In their paper, Alavi and Bar-Joseph adapt an image analysis method (iterative closest point) used for point-set registration to the context of single-cell RNA-seq. Specifically, the method provides an alternative technique for ‘aligning’ different scRNA-seq datasets, generally for the purpose of batch correction. Batch correction is an important problem in scRNA-seq because of the diversity of methods used to collect the data and the potential power of integrating across many datasets, and several batch correction methods have been proposed in recent years. The new method, SCIPR presented here appears to outperform existing methods in the scenarios tested and, importantly, SCIPR provides a transformation in gene-space, so that transformed data can be used in downstream analyses. Overall, this paper is clearly written and the new method will be useful, however certain choices regarding controls and test datasets (as detailed below) should be examined more carefully before publication.

Major

1. My primary concern regarding the benchmarking analyses presented in the first part of the methods has to do with choices in test datasets. First, the primary dataset that was analyzed (CellBench), while clearly useful for benchmarking, does not represent the usual complexity of single-cell RNA-seq datasets. More representative would be the pancreas dataset used. I would like to see the tSNE embeddings from the pancreas presented as well (similar to Figure 4).

2. In addition, it concerns me that only highly variable genes were used during the analysis. While perhaps this was done to decrease the computational complexity, it will also decrease the usefulness of the transformation if only a small subset of genes can be used in downstream analyses. This limitation should be discussed and perhaps compared to a dataset filtered only on depth. In addition, I believe these filtering details should be included in the methods or results section as they are important to evaluating the method.

3. I am skeptical of the analysis presented in section 2.5. It is not obvious to me that you would expect the coefficients for batch effect correction to be relevant to the biological process. Would it not be more likely that the genes with the highest coefficients would be related to technical variation? (stress perhaps?)

4. Also on section 2.5, I don’t believe proper controls were demonstrated in this analysis – we would expect highly variable genes to already be enriched in the relevant biological process. The authors compare the high coefficient genes selected by SCIPR to a background of all genes filtered by depth, including low variable genes that were not included in the SCIPR analysis. The authors should either use only the set of highly-variable genes as background, or provide a null analysis comparing a random subset of highly-variable genes to all genes filtered by depth. I believe these methods details should be included in the methods section and not in the supplement as they are crucial to interpretation.

Minor

Line 32 – I believe some of the largest studies are combinatorial indexing based (split-seq/sci-rna-seq) rather than solely microfluidic

I found the iLISI/cLISI boxplots difficult to evaluate ‘at a glance’. I think the visualization might be improved if overlaid by a line plot connecting the median values.

I was able to successfully install the software using pip and links to data analysis on Github worked.

Reviewer #2: Inspired by the point-cloud registration problem in computer vision and robotics, the authors propose a novel algorithm named SCIPR for alignment of scRNA-Seq datasets from multiple experiments. Experimental results on three benchmark scRNA-Seq datasets reveal that SCIPR not only achieves better accuracy in alignment using original datasets, but also can generalize to unseen datasets. Moreover, the alignment results of SCIPR can derive useful insights into the selection of important genes (SCIPR genes). Finally, gene ontology (GO) enrichment analysis demonstrates that SCIPR genes are overlapped with more GO terms than differentially expressed genes.

However, there are a couple of issues that would limit the possible impact of the work, which I summarize below.

Scientific issues:

1. Since the authors developed two versions of SCIPR based on Mutual Nearest Neighbors (MNN) and greedy matching respectively, the authors are recommended to specify the concrete situations that each of the two algorithms is applicable.

2. In Line 66-73, why can SCIPR (or ICP) enable the maintenance of semantics and generalization is not well addressed.

3. To demonstrate the convergence rate of SCIPR, it is necessary to add a table or figure to illustrate the convergence rate on datasets of different sizes.

4. Because the final global transformation function is the composition of functions learnt in each iteration, it seems that the more iterations, the more parameters the global transformation function has. Will this lead to overfitting?

5. In Figure 2, there are indrop1, 2, 3, and 4, but only indrop 1 and 3 are mentioned in Line 98-99. Then in Methods section (Line 206), the author only showed data information for indrop3 and 4. Also, the word "indrop" should be "inDrop".

6. In Table 1, Nine methods are presented. Why only four algorithms are selected for comparison? Besides, what does “none” represent in Figure 2 ~ 4?

7. In Figure 4, I can hardly tell which one is better between SCIPR-gdy and MNN.

8. The description of LISI, iLISI, and cLISI is less informative. For people not familiar with this score, it is hard to follow. Detailed algorithms are preferred to better explain these scores, as they are the most important validation method in this paper.

9. The paper showed integration of two scRNA-Seq datasets from different experiments/technologies, while in many cases multiple sample alignment is needed. How could SCIPR deal with such situation, align two datasets first and apply the third one, or align all three simutaneously? If it is the first situation, does the alignment sequence (e.g., 1+2+3 and 1+3+2) matter?

10. In Section 2.5, the selected genes are selected according to their SCIPR coefficients. Why it is written that the parameters learnt can be used to identify CELL TYPE-SPECIFIC genes?

11. In section 2.5, I would prefer a toy case to show the comparison of DE results. Does SCIPR find more biological meaningful DE genes than other tools, such as Seurat? Stating the DE function and only using GO enrichment are insufficient to support the title of section 2.5.

12. As known, the number of GO terms overlapped with a gene set can be impacted by the size of the gene set and the size of the GO terms. Please present detailed information including the size of the gene set and the size of the GO term.

13. It is necessary to add a comparison of efficiency between the methods.

14. A discussion should be added to explain the choice of parameters alpha, beta, and k.

Presentation issues:

1. A number of typos exist, for example, “non of the current methods …” in line 66.

2. The tables and figures can be further polished.

Reviewer #3: The authors developed Single Cell Iterative Point set Registration (SCIPR), which extends methods successfully applied to align image data to scRNA-Seq. The idea is novel and interesting. The method is technically sound. The iterative approach is nice. The comparison with other methods is mostly convincing, although it could be more thorough and more rigorous. Although it requires much more tests before a tool like this could dominate real applications, it seems promising. The code is available to the public, which provides a good opportunity for others to repeat this work and test on more data.

One of the major issues in scRNA-Seq is that the dropout rate is high (missing values of gene expression). It is unclear how the authors handled it. Did they use imputation? Some details should be provided.

The authors should discuss some limitations of their work. For example, the method uses anaffine transforms, where non-linear properties are not considered.

The writing of this paper can be improved. It is a little hard to follow. It may benefit from moving some text on problem formulation to Introduction or Results. Some discussions are sketchy. For example, in Figure 4, the Seurat's performance is also very good, but it was not mentioned.

Figure quality needs some improvement. For example, A and B should be labelled in Fig. 5 (2).

The language needs some edits, e.g.:

non of the current methods  none of the current methods

...

**Have all data underlying the figures and results presented in the manuscript been provided?**

Reviewer #1: Yes

Reviewer #2: Yes

Reviewer #3: Yes

PLOS authors have the option to publish the peer review history of their article (what does this mean?). If published, this will include your full peer review and any attached files.

Reviewer #1: No

Reviewer #2: No

Reviewer #3: Yes: Dong Xu
---

## [Decision Letter · Decision Letter 1]

18 Aug 2020

Dear Dr. Bar-Joseph,

We are pleased to inform you that your manuscript 'Iterative point set registration for aligning scRNA-seq data' has been provisionally accepted for publication in PLOS Computational Biology.

Best regards,

Shi-Jie Chen

Associate Editor

PLOS Computational Biology

Weixiong Zhang

Deputy Editor

PLOS Computational Biology

Reviewer's Responses to Questions

**Comments to the Authors:**

Reviewer #1: I am satisfied that the authors have adequately addressed my concerns and recommend publication.

Reviewer #2: The authors have fixed all my issues.

Reviewer #3: The authors addressed reviewers’ comments well. The revised version is improved in quality. I have no further suggestions to make.

**Have all data underlying the figures and results presented in the manuscript been provided?**

Reviewer #1: Yes

Reviewer #2: Yes

Reviewer #3: Yes

PLOS authors have the option to publish the peer review history of their article (what does this mean?). If published, this will include your full peer review and any attached files.

Reviewer #1: No

Reviewer #2: No

Reviewer #3: **Yes: **Dong Xu

---

## [Editor Report · Acceptance letter]

19 Oct 2020

PCOMPBIOL-D-20-00745R1 

Iterative point set registration for aligning scRNA-seq data

Dear Dr Bar-Joseph,

I am pleased to inform you that your manuscript has been formally accepted for publication in PLOS Computational Biology. Your manuscript is now with our production department and you will be notified of the publication date in due course.

With kind regards,

Matt Lyles
